# The Underrated Gut Microbiota Helminths, Bacteriophages, Fungi, and Archaea

**DOI:** 10.3390/life13081765

**Published:** 2023-08-18

**Authors:** Maria Jose Garcia-Bonete, Anandi Rajan, Francesco Suriano, Elena Layunta

**Affiliations:** 1Department of Medical Biochemistry and Cell Biology, University of Gothenburg, SE-405 30 Gothenburg, Sweden; 2Instituto de Investigación Sanitaria de Aragón (IIS Aragón), 50009 Zaragoza, Spain

**Keywords:** microbiota, gut–microbiota interactions, intestinal diseases, dysbiosis, intestine, therapeutic strategies

## Abstract

The microbiota inhabits the gastrointestinal tract, providing essential capacities to the host. The microbiota is a crucial factor in intestinal health and regulates intestinal physiology. However, microbiota disturbances, named dysbiosis, can disrupt intestinal homeostasis, leading to the development of diseases. Classically, the microbiota has been referred to as bacteria, though other organisms form this complex group, including viruses, archaea, and eukaryotes such as fungi and protozoa. This review aims to clarify the role of helminths, bacteriophages, fungi, and archaea in intestinal homeostasis and diseases, their interaction with bacteria, and their use as therapeutic targets in intestinal maladies.

## 1. Introduction

The gastrointestinal (GI) tract is the habitat of a complex and abundant population of organisms including bacteria, viruses, archaea, and eukaryotes (e.g., fungi and protozoa), collectively known as the gut microbiota. The microbiota refers to the living organisms present, while the term microbiome goes beyond the microorganisms themselves and considers their structural elements, genomes, metabolites, interactions, and the surrounding environmental conditions [1]. The microbiota is considered a critical factor in the maintenance of intestinal health [2]. Previous studies have indicated that the gut microbiota modulates the whole intestinal physiology, including digestion, absorption, and secretion [3], as well as motility [4] and defense [5,6], thereby being considered a crucial player in human health and disease [7].

In particular, the importance of the microbiota in intestinal health is related to its implication in vitamin synthesis, regulation of the gut barrier function, modulation of the immune system, digestion of nutrients, and the protection against opportunistic pathogens [8,9]. However, gut microbiota deviations, termed as dysbiosis, not only favor intestinal infection by pathogens [10] but are also linked with many diseases, including lung diseases, cardiovascular diseases, cancer, obesity, type 2 diabetes, and intestinal bowel diseases (IBD), among others. While the role of bacteria is widely defined, there is a knowledge gap concerning the influence of unusual gut microbiota such as helminths, bacteriophages, fungi, and archaea in intestinal balance and their role in GI diseases.

Bacteria are the main microorganisms studied from the gut microbiota because they represent the majority of the microorganisms in the gut (>90%), in addition to their relative ease of culture and isolation. Until the last two decades, the techniques to identify and study new microorganisms were focused on culture-based methods, light microscopy, staining, and biochemical assays [11,12]. It was not until the mid-2000s that high-throughput low-priced sequencing enabled us to analyze more complex samples (e.g., human stool), sequence whole genomes, and identify new species. Using 16S and 18S rRNA sequencing allowed us to identify other microorganisms present in the gut microbiota that could not be identified for a long time, such as archaea and many fungi, apart from many new bacteria that have never been cultured before.

Other techniques, such as Internal Transcribed Spacer (ITS) [13] and real-time qPCR of the mcrA gene [14,15], have also been vital for the identification of new fungi and methanogens in the microbiota, respectively. Nucleic acid and immunological detection are the most common methods used for virus identification [16]. The lack of 16S/18S subunits in viruses makes identifying unknown viruses more difficult using high-throughput (HTS) techniques. However, new HTS and metagenomics approaches have been developed for viral identification in the last few years [17,18,19].

In the current review, we focus on the role of the unusual microbiome in intestinal homeostasis and its interaction with the intestinal mucosa. Moreover, we summarize the latest updates in medical treatments based on the modulation of helminths, bacteriophages, fungi, and archaea (Figure 1) as a therapeutic approach for GI-related maladies, such as IBD, colitis-associated cancer (CAC), or metabolic syndrome, among others.

## 2. Helminths

Helminths are considered one of the most important infectious agents in the GI tract, causing severe morbidity worldwide, especially during early life, with more than 1.5 billion individuals affected annually [20]. In this context, some helminths, such as the phylum Acanthocephala, can infect the intestines due to their proboscises being decorated with spines that are used to attach to the intestinal epithelial cells during infection [21]. Moreover, helminths such as Acanthocephala can infect many animals, including invertebrates, fish, amphibians, birds, and mammals [22]. Therefore, helminths’ efficiency in colonizing and infecting the GI tract is due to their ancient origin and their co-evolution with humans [23]. This theory can be explained from an immunity perspective since individuals have shaped a typical anti-helminth immune response due to the similarities in the immunobiology of different helminths [24].

The most prevalent helminth phyla are Cestodes (known as tapeworms), Nematodes (known as roundworms), and Trematodes (known as flukes) (Table 1).

Helminths have robust immunoregulatory activity in the host and can modify intestinal microbiota composition, causing dysbiosis [25]. Helminths can modify the gut microbiota and the production of their metabolites, such as short-chain fatty acids (SCFAs), which could promote an ongoing helminth infection to the detriment of the commensal microbiota [26]. However, this phenomenon seems to depend on the helminth type, and its specific mechanism is still unknown [27]. Furthermore, helminths can secrete C-type lectin domain-containing proteins, which are antimicrobial products that agglutinate bacteria and reduce microbiota composition [28], and peroxiredoxin that neutralizes the reactive oxygen species (ROS) released from the host [29] in order to reduce commensal bacteria that inhibit helminth infection. Beneficial intestinal microbiota can also inhibit helminth gut colonization by different mechanisms [30]. For example, microbiota can regulate intestinal motility to favor contractile expulsion of a parasite and avoid helminth infection in the gut [31]. In this context, an extensive description of this interaction between helminths and bacterial microbiota is detailed in the work from Llinás-Caballero K et al. [32].

Intestinal microbiota may improve, in some cases, the colonization by helminths. A recent study indicates that infection by *Heligmosomoides polygyrus* in germ-free (GF) mice may be reduced, suggesting that bacteria regulate immune pathways that benefit the parasites [28]. In other cases, the eggs of some parasites, such as *Trichuris muris*, need to be in direct contact with intestinal bacteria to hatch and continue their infective life cycle [33]. However, intestinal microbiota can trigger the opposite effect of helminths.

Several works have described how intestinal microbiota, such as *Akkermansia muciniphila*, can protect the intestinal epithelium from helminths since it promotes the production of intestinal mucus that separates these parasites from the intestinal epithelial cells [34]. In addition, a study carried out by Moyat M et al. showed that in most cases, the intestinal microbiota provides parasite infection resistance since mice lacking a complex bacterial microbiota exhibited decreased levels of acetylcholine in the gut and, therefore, reduced intestinal motility [31]. Nonetheless, other studies describe that the helminth–diabetes comorbidity (which is characterized by an impaired gut barrier) could reduce the systemic inflammation and microbial translocation observed in type 2 diabetes mellitus [35]. In this context, another scientific group has proved that SPRR2A, a protein that may damage negatively charged bacteria, is induced by helminths to avoid pathogenic bacteria translocation across the intestinal epithelium [36]. Therefore, intestinal microbiota and helminths have an intense crosstalk relationship whose result may depend on both the bacteria type and the helminth phylum.

Helminths have rarely been reported as a cause for chronic gastrointestinal diseases such as IBD or CAC [31], yet in most cases, their etiopathology is still unknown [37]. Recent therapeutic strategies in IBD and CAC have suggested using helminths to ameliorate inflammation and improve the quality of life in IBD and CAC patients [38,39]. These innovative therapies are based on epidemic studies where a high prevalence of infection with helminths during early life may protect against IBD in adulthood [40], which suggests imprinting an intestinal defense not just by bacterial microbiota during early life [41] but also by helminths. Pre-treatment with different helminth species such as *Echinococcus multilocularis* (attenuation of T helper Type 1/Type 17-mediated immune response), *Heligmosomoides polygyrus* (activation of colonic Foxp3+ T cells), or *Hymenolepis diminuta* (increased propensity for T helper-2 immunity) seems to reduce inflammation in chemically induced colitis in mice [42]. However, this protection from gastrointestinal inflammation may depend on the parasite used [43], thereby underlying the importance of certain helminths for our health.

Generally, helminth infection primes a strong Th2 immune response in the gut, triggering the secretion of interleukins (IL) such as IL4 and IL10, among others. In addition, these parasites cause hyperplasia of mast cells, eosinophils, goblet cells, and innate lymphoid cells, as well as the expansion of regulatory T cells (Tregs) [44]. Thus, treating an ongoing intestinal inflammation with helminths may involve the upregulation of anti-inflammatory cytokines such as IL10, the anti-inflammatory transforming growth factor (TFG)-β, and the Treg cell recruitment that prevents inflammation [45]. Recent researchers have shown that administering a single serine protease from *Trichinella spiralis* prior to colitis ameliorates gut inflammation, reduces intestinal epithelial damage, and decreases the release of pro-inflammatory factors [42].

The benefits of helminth therapy prior to intestinal inflammation would not be limited to the immunomodulatory effect of these organisms in the host. Previous works have indicated that helminths may protect from intestinal disturbances by blocking colonization by inflammatory *Bacteroides* species [46]. Supporting these data, individuals in helminth-endemic regions possess a protective microbiota against inflammatory bacteria, and a de-worming treatment induces the reduction in protective Clostridiales and the increase in pro-inflammatory *Bacteroides* species. Thus, the benefits of helminths in the prevention of intestinal inflammation would not only involve immunomodulation of the GI tract but also blocking gut colonization by harmful microorganisms [46].

Further, an infection with helminths would not be limited to a harmful effect but may also help develop therapies for intestinal inflammation and associated cancer. Moreover, intestinal immunity may not always react to helminth infection or may elicit a mild defense response, suggesting that tolerance depends on the helminth species and abundance. In this context, previous works raise the question of whether increased hygiene could lead to the disappearance of helminths that prime intestinal immunity in early life, triggering the worldwide increase in diseases such as asthma, IBD, or type 1 diabetes, among others [47], which is known as the “hygiene hypothesis” [48].

Novel data highlight the importance of environmental factors in developing effective and strong intestinal immunity during early life to protect individuals during childhood and their whole lifespan [49]. During this period, known as the “window of opportunity”, not only is the commensal microbiota critical [50], but also the opportunistic infections by helminths. Hence, this period is not only a window of opportunity for the development of intestinal immunity but also a “window of vulnerability” that can disrupt the regular development of intestinal defense.

Infection by helminths during early life seems beneficial since it would prime the development of type 2 immune responses, especially type 2 innate lymphoid cells (ILC2) that trigger the synthesis of anti-inflammatory M2 macrophages during childhood and help avoid immune hyper reactions during adulthood [51]. Helminths boost the adaptive T helper 2 response in childhood that would be maintained during adulthood without higher sensitivity to allergens, suggesting that punctual helminth infection may improve immunity development [52]. However, other researchers discard this idea and highlight that helminth infection may disturb bacterial microbiota in early life and the normal development of immunity in the gut [53].

In conclusion, helminths are ancestor microorganisms whose infective mechanisms have co-evolved with the host. These organisms compete with commensal microbes and are critical immunoregulators in the intestine, where their impact on the development of intestinal immunity during early life is crucial. Finally, helminths may be used soon as new tools for IBD therapy since they ameliorate intestinal inflammation. However, their use is still controversial and may depend on the organism selected as well as the specific characteristics of the patient, including genetics, nutrition, and lifestyle.

## 3. Bacteriophages

Several recent studies have shown that bacteriophages (viruses that require bacteria as their host for infection and replication) play an important role within the GI tract. This section focuses on bacteriophages as crucial components of GI physiology.

Bacteriophages were discovered in 1915 and 1917 [54] when it was observed that they could lyse a lawn of bacterial growth and be cultivated as an infectious agent. The International Committee on the Taxonomy of Viruses (ICTV) has classified bacteriophages into ten families based on their structure and genome (Table 2), which include capsid structure, the presence or absence of a tail, and genomic nucleic acid type [55]. There are 1031 bacteriophages (also referred to as phages) in total, making them the most abundant biological units in the world.

Based on their infectious cycle, phages are classified as lytic or lysogenic. Lytic phages, such as *Escherichia coli*’s T4 phage, first attach to a receptor on their host and then release their genomic DNA into the host’s cytoplasm. That is followed by the replication of the phage genome, expression of phage proteins, assembly of progeny, and release of new virions, all of which take place using host cell machinery. By contrast, lysogenic phages, such as *E. coli*’s λ phage, integrate into the host chromosome or form a linear or circular self-replicating plasmid in the host cytoplasm, a prophage, after initial attachment and genome injection. A bacteria can host a prophage continuously while going through its replication cycles and the bacteria can be unaffected by the prophage or can undergo changes in its resistance to other phages or in its pathogenicity [56].

Bacteriophages cannot propagate without bacteria, and thus, the human gut, with trillions of bacteria, is a perfect environment for phages to thrive. Among all the viral genomes in the human gut, phages comprise a majority of 97.7% [57]. However, very little is known about the diversity of the gut phageome [58,59,60] since phages are challenging to cultivate and phage genomes are very variable and are difficult targets to analyze. Metagenomic data on fecal samples revealed that approximately 90% of gut phages are unclassified, but the remaining 10% mainly belong to the order Caudovirales (non-enveloped, tailed, icosahedral capsid with dsDNA; *Podoviridae*, *Myoviridae*, and *Sidoviridae*) followed by the families *Microviridae* (non-enveloped, non-tailed, icosahedral capsid with circular ssDNA) and *Inoviridae* (non-enveloped, non-tailed, filamentous with circular ssDNA). The two most abundant phages from the order Caudovirales and family *Podoviridae* are crAssphage (cross assembly phages; discovered in 2014 by **cr**oss **A**ssembly reads from a metagenomics analysis of fecal samples) [61] and Gubaphage (gut Bacteroidales phage) [62] and the genus *Bacteroides* serve as their hosts. CrAssphages are omnipresent and plentiful in the human gut microbiome and, due to this ubiquity, are even utilized as a marker for human feces [63].

Bacterial dysbiosis has been strongly correlated with GI disorders. Many studies have reported the same occurrence with the gut phageome. Studies in humans [64,65,66,67,68] have shown an alteration in the gut phageome in patients with Crohn’s disease (CD), ulcerative colitis (UC), irritable bowel syndrome (IBS), colorectal cancer (CRC), celiac disease, and environmental enteric dysfunction (EED) (Table 3).

A *Microviridae*-rich phageome in a healthy colon switches to a phageome-rich one in different members of Caudovirales in patients with IBD. Since bacteriophages require bacteria to replicate, the phageome composition reflects the bacterial composition. IBD patients were shown to have lower levels of Firmicutes bacteria and more Firmicutes phages than healthy controls [67]. Along with healthy and disease states and dietary components, age is also a key regulator of gut phageome composition. There are few to no phages in newborn stool samples, but the number increases dramatically after the first month of life [70]. When there are low numbers of bacteria in the gut in early life, the bacteriophage population consists mainly of lytic phages. These lytic phages are procured as prophages through vertical transmission from the mother or derived from the environment. A scarce number of bacteria and phages at this stage of life does not allow for sustained levels of lysogenic phages, but as bacterial diversity increases, lysogenic phages predominate and stabilize the viral gut population. Prophages modify cell surface receptors and proteins and prevent secondary or co-infections of their host. Lytic Caudovirales predominate the phage composition in early life, and their amount declines by three years of age, while lysogenic *Microviridae* is present in low amounts in neonates and infants, but they become more abundant in two- or three-year-old children.

After this age, the phageome stabilizes and has mostly the same composition throughout life [57,70,71,72,73]. A few studies have shown that the first phages to colonize an infant’s gut originated from Bacteroidetes, Firmicutes, and Actinobacteria, i.e., the first bacterial colonizers, where the phages were present as prophages and were transferred to the child during breastfeeding [70,74]. Phages belonging to Caudovirales were shown to be transmitted from mother to child as prophages in Bifidobacteria [67,75]. In early life, Caudovirales make up most of the phageome but contain crAssphages only in small numbers. CrAssphages appear when *Bacteroides* begin colonizing the gut (between one and three months of age) and then become the most abundant component of the phageome in adult life [76]. Thus, bacterial dysbiosis is correlated with an imbalance in the phageome, and in turn, this is associated with GI disorders.

Bacteriophages constitute many gut viral genomes and are good candidates for modulating human immunity. In vivo and clinical studies [77,78] have shown phages to modulate immunity by inducing both pro- and anti-inflammatory pathways. On the one hand, *Staphylococcus aureus* and *Pseudomonas aeruginosa* activate the transcription of IL-1, IL-6, and TNF-α when incubated with peripheral blood monocytes; endotoxin-free *E. coli* induces the production of IFN-γ and *Lactobacillus* and *Bacteroides* aggravate colitis via IFN-γ through toll-like receptor 9. On the other hand, T4 phages could reduce ROS caused by bacterial infections [79], inhibit NF-κB activation, and, in turn, hinder infections by pathogenic viruses [80].

Phages are vital elements of the human GI tract and largely contribute to the dynamics of the gut ecosystem. An aspect of the phageome that could open new avenues in GI disease therapeutics is to develop bacteriophages as weapons against bacterial diseases. Phage therapy is a promising strategy, and it can one day eliminate challenges with antimicrobial resistance. When Felix d’Herelle coined the term “bacteriophage” in 1917, he started treating patients with oral doses of a *Shigella*’s phage [81]. He also treated cholera patients with *Vibrio cholerae*’s phages [82]. Duan and coworkers [82] provide a proper review of clinical trials and case reports examining phage therapy ranging from pancreatitis [83] to Crohn’s disease [84]. Phages are considered safe as a treatment option due to their specific host range, but it is still not known what effect their prolonged presence will cause and if they may start interacting with human cells. Having a specific host also means that phage therapy cannot be used as a broad-range therapeutic. This approach requires a clear understanding of the gut phageome and bacterial microbiome so that it can be fully utilized to benefit human health.

## 4. Fungi

In recent years, there has been an increased interest in the gut mycobiota, which is collectively called intestinal fungi. As a fundamental part of the human microbiome, mycobiota are broadly distributed in the healthy human body, primarily on the skin and nearly all the mucosal surfaces, like the GI tract, oral cavity, skin, and vagina [85]. Targeted and shotgun high-throughput sequencing technologies have allowed us to investigate deeper and understand microbial diversity at various locations. In the gut, for example, recent shotgun sequencing efforts have suggested that fungi make up approximately 0.1% of the total microorganisms [86]. However, there are several reasons (e.g., under-detection compared with bacteria in shotgun sequencing efforts, differences in biomass and genomic size) to question whether this underestimates fungi’s number and significance [86,87]. Generally, approximately 70 genera and more than 184 species of mycobiota colonize the human gut, with *Candida*, *Saccharomyces*, and *Cladosporium* species being dominant [85].

Despite their relatively small number in the gut community, fungi are essential in our health since they are implicated in the regulation of several functions (e.g., nutrition, metabolism, and immunity), both at the intestinal and extra-intestinal (e.g., lung, liver, and brain) levels [88,89]. Thus, fungi are associated with the health and disease state of the host [88,89]. This interaction between fungi and host can be seen as a spectrum of symbiotic relationships (i.e., commensal, parasitic, mutualistic, and amensalism) [90]. For example, the yeast *Saccharomyces boulardii* is widely used as an effective probiotic to prevent and treat pathogenic bacterial infections and acute/chronic intestinal complications (e.g., *Clostridium difficile* and *Helicobacter pylori* infections, diarrhea, and IBD) [91,92]. Many mechanisms of action have been attributed to *S. boulardii*, including its ability to re-establish the intestinal microbiota homeostasis, interfere with the ability of pathogens to colonize and infect the mucosa, regulate the local and systemic immune response, and induce enzymatic activities promoting absorption and nutrition [92]. Additionally, it is crucial to point out that the effect that members of the gut mycobiota exert on the host is also closely linked to targeted bacteria and to the respective production of primary and secondary metabolites with biological and anti- or pro-microbial activities [93,94]. That underlines the presence of functional connections between bacterial and fungal communities in the gut, which are further discussed below.

Like the bacterial microbiota, it is also noteworthy to mention that multiple host-related factors, such as diet, age, and sex, are implicated in the modulation of the gut mycobiota composition [95,96]. Fungal microbiota deviations are often associated with the progression of certain intestinal (e.g., IBD, celiac disease, and colon cancer) and extra-intestinal diseases (e.g., pulmonary infection, hepatic cirrhosis, and multiple sclerosis) [88].

This section will mainly emphasize the importance of gut mycobiota dysbiosis and bacterial–fungal interactions in intestinal inflammation (i.e., IBD) and fungal infections.

Sokol et al. have shown a distinct fungal microbiota dysbiosis in IBD patients characterized by alterations in biodiversity and composition (i.e., increased Basidiomycota/Ascomycota ratio, a decreased proportion of *Saccharomyces cerevisiae*, and an increased proportion of *Candida albicans*) [97]. They also emphasized a disease-specific inter-kingdom network alteration in IBD, thereby highlighting that beyond bacteria, fungi might also play a pivotal role in IBD pathogenesis [97]. Along with this study, an increased fungal load and a divergent mucosa-associated fungal microbiota characterized by an increase in species with potential pro-inflammatory effects (i.e., Xylariales) and a decrease in species with potential beneficial effects (i.e., *Saccharomyces cerevisiae* and *Filobasidium uniguttulatum*) were observed in CD patients [98].

Another common yeast, *Malassezia restricta*, is recognized in most patients carrying the IBD risk allele caspase recruitment domain-containing protein 9 (CARD9), a molecule implicated in fungal innate immunity [99]. However, how fungi interact with the host immune system is still not well known. Generally, fungi and fungal microbial-associated molecular patterns (MAMPs) are mainly associated with fungal cell walls that can interact with membrane-bound receptors, such as lectin receptors, toll-like receptors (TLRs), and scavenger receptor family members [86,100]. This interaction leads to the activation of many mechanisms of defense (e.g., phagocytosis, respiratory burst, activation of transcription factors, and production of pro-inflammatory cytokines and chemokines), which promotes host defense against fungi and helps to reduce increased susceptibility to fungal infections [86,100].

Among the different types of cytokines produced by the immune system, IL-17 and IL-22 are considered necessary for host defense against fungi. Genetic mutations in each of these molecules are linked to increased susceptibility to fungal infections in humans [86]. Fungi can also interact with bacteria, and it has been shown that commensal fungi in the mouse gut can be found in patches together with gut bacteria [101]. Studies in GF and antibiotic-treated mice (i.e., short- and long-term treatment) have also demonstrated that mice are highly susceptible to *Candida* infection and colonization by and the overgrowth of *Candida* species in the mouse gut, respectively [101,102,103,104]. The same observations were underlined in humans, where extended antibiotic treatment can increase the susceptibility to fungal infections, primarily ascribed to the expansion of *Candida* species [105,106]. Keeping with the idea to demonstrate the causal role of fungi in the modulation of bacterial composition, the work from Bernardes et al. [107] demonstrated that the colonization of GF mice with a defined consortia of either bacteria, fungi, or both elicited robust microbiome and immunological shifts that modulated the subsequent susceptibility to mucosal inflammation in the distal gut and the lung. These results suggested a synergistic inter-kingdom relationship between bacteria and fungi, which has been described in other ecosystems [108].

Along with this study, the introduction of *Candida albicans* into antibiotic-treated mice was associated with disturbances in bacterial community reassembly, further pinpointing how certain fungi in numerically inferior numbers in the bacterial microbiome can drive substantial bacterial microbiome deviations [109]. Furthermore, it is critical to point out that despite the different infections caused by *Candida* species, this fungus in the presence of an equilibrate/diverse bacterial community and of specific host conditions can improve human health [110]. Indeed, as shown both in humans and animal studies, *Candida albicans* contributes to training our immune system by shaping T-cell responses, competing with other pathogens, and influencing the balance of the intestinal microbiota [110]. Thus, these findings highlighted that within the gut microbiome there is a broad array of ecological interactions between fungi and bacteria, and that the intensive fungal–host–microbiota crosstalk is of utmost importance for our health.

## 5. Archaea

Archaea is a group of single-cell microorganisms representing life’s lesser-studied domain. These microorganisms are prokaryotic and were considered archaebacterial or extremophilic bacteria (microorganisms that live under extreme conditions) until 1978 [111]. Currently, there is more robust evidence that archaea are evolutionarily distant from bacteria. Indeed, they differ from the other domains by their cell structure (they lack peptidoglycan in the cell wall and instead have a protein coat), their metabolism, and their molecular machinery. One of the main differences between archaea and the other two domains of life is the type of lipids present in their membranes. Their membranes are not composed of glycerol ester or fatty acids like bacteria and eukarya membranes, respectively. They are formed by isoprenoid chains ether-linked to glycerol-1-phosphate (G1P) [112,113]. These lipids allow archaea to form monolayers in addition to bilayer membranes, such as bacteria and eukaryotes [114], providing them with other advantages to survive extreme conditions [115,116]. The domain Archaea includes a wide variety of organisms that share properties with both bacteria (various morphologies: coccus, spirillum, bacillus, or irregular shapes; have a single circular chromosome) and eukaryotes (similar molecular machinery for DNA replication, RNA transcription, and protein translation and the presence of histones for chromosomal DNA packaging) [117]. They have been found in many different extreme environments, and they have been classified as hyperthermophiles (high-temperature conditions), piezophiles (high pressure), acidophiles and alkaliphiles (low and high pH), halophiles (high salt concentrations), and methanogens (living in anaerobic conditions and producing methane as a metabolite). Archaea display many lifestyles, including anaerobic and aerobic respiration, fermentation, chemoautotrophy, heterotrophy, and photoheterotrophy, allowing them to colonize various environments. Some mesophilic archaea live in less hostile environments and have been found as part of the human microbiota.

Thanks to the advances in genomic sequencing and computational approaches, the number of identified archaea genomes has increased considerably in the last decade. That has allowed us to classify them into three major superphyla, Asgard, DPANN, and TACK [118], and the Euryarchaeota phylum, which does not correspond to any of the superphyla and comprises the archaeal species found in the human microbiota. Due to the difficulties in culturing and identifying archaea, our knowledge is minimal, and extensive studies must be conducted.

Archaea are everywhere, and although they have not been associated with pathogenesis, several studies have shown certain relations with human diseases [119]. They have been the forgotten players in the study of the microbiome [120]. However, in the latest decade, several species have been isolated from human skin and the nose, lung, oral cavity, vagina, and GI tract [121,122], pointing to the importance of considering these microorganisms in future studies of the human microbiota and their possible implication in diseases. Archaea represent a small proportion of the total gut microbiota [123,124,125], but their implication in human health and gut diseases has become an important research topic lately.

Methanogens are the predominant archaea group identified in the human gut microbiota [9]. That is unsurprising considering that this type of archaea is related to the degradation of carbon products (such as H_2_-CO_2_, formate, acetate, and methanol) to methane and has been identified before in ruminating animal’s intestines [126]. Methanogens can use the hydrogen generated from bacteria fermentation and promote the host’s digestion, transform heavy metals, and use the trimethylamine produced by intestinal bacteria, reducing the risk of cardiovascular diseases [127]. In fact, these relations with bacteria suggest a symbiotic metabolism.

Methanobacteriales and Methanomassiliicoccales are the most abundant orders of archaeal methanogens in the human gut [123]. *Methanobrevibacter smithii* is the most identified archaea in the human gut, representing more than 95% of the gut archaea in healthy individuals [128] and acting as the primary producer of methane in the gut. *Methanosphaera stadtmanae* and *Methanomassiliicoccus luminyensis* are other species that have been commonly isolated from the gut, but they require more restricted nutrients and conditions to grow than *M. smitthi*. Changes in the prevalence of *M. stadtmanae* in the gut have been related to several diseases, which we have described further. Although these three species represent the majority of archaea identified in the human microbiota, other methanogens, halophilic archaea, and members of the orders Sulfolobales [129] (phylum Crenarachaeotas) and Nitrososphaerales [130] (phylum Thaumarchaeota) have also been found.

On the one hand, the decrease in methanogens in the gut has been related to CD, UC, and malnutrition [127]. A malnutrition study in children in Senegal and Niger showed an increase in fecal redox potential, a decrease in total bacteria, and a dramatic depletion of *M. smithii* [131]. The reduction in methanogens and methane production in the gut has also been observed in UC and CD patients [15,132]. On the other hand, the increase in methanogens or methane production has been related to diverticulosis, IBD, irritable bowel syndrome (IBS), constipation [133,134], obesity, CRC, and polyposis [127].

A metagenomics study [135] in more than five hundred patients including colorectal cancer, adenomas, or healthy controls showed an enrichment of halophilic archaea and a depletion of methanogens in the stools of patients with CRC. A progressive increase in the prevalence of *Natrinema* sp. J7-2 (halophile) was observed from individual controls compared to patients with adenoma and to patients with CRC. This study also identified an association between the bacteria and archaea species in healthy individuals, and reported that this association is lost in patients with CRC. Halophiles and methanogens, including *M. stadmanae*, were increased in individuals above 50 years of age, and the abundance of *M. smithii* was significantly decreased (Table 4).

A study with stool samples of children between six to ten years of age in The Netherlands suggests the implication of archaea in the dysbiosis–asthma paradigm [136,137]. According to Barnett et at., 2019 [136], *M. stadtmanae* in the stool samples was associated with a lower risk of asthma independently of the parental asthma status. They observed a tendency of lower eczema, aeroallergen, and food allergen sensitization in the presence of this archaeon [136]. However, the presence of *M. smitthi* was not associated with the incidence of asthma; this study speculates the possibility of tolerogenic effects by early-life archaeal exposure [136].

Immunological studies have demonstrated the immunogenic properties of methanoarchaea in human cells and mice [138,139]. The human innate immune system specifically recognizes *M. stadtmanae* or *M. smithii* and can induce an inflammatory cytokine response. Furthermore, *M. stadtmanae* induces a much stronger inflammatory reaction than *M. smithii* [138,139]. In addition, a bioinformatics study predicted the allergenicity of different proteins present in archaea and bacteria and suggested that these pro-inflammatory proteins are predicted more often among archaea than bacteria and that most of these proteins were among the secreted proteins [139,140]. That suggests that although archaea are not considered pathogenic, they may contribute to pathogenic conditions of the human GI tract, and further investigation is required to assess their involvement.

Differences in the gut archaeome have been observed with the incidence of obesity. An increase in the orders of Methanosarcinales and Methanococcales and a decrease in the orders Natrialbales, Methanocellales, and Thermoproteales were observed in obese patients compared to lean control patients [141]. Furthermore, the increased methane detected in the breath was correlated with overweight subjects and higher body mass index [142].

An extensive study of more than eight hundred Korean subjects [123] reported that the abundance of the human gut archaea is approximately 10–14% of the total bacteria/archaea abundance in the gut. Although the majority of the archaea identified in the Korean archaeome were from the methanogen group (Metabobacteriales: *M.smithii* and *M.stadmanae*), the study also identified some haloarchaea genera (*Halolamina*, *Haloplanus*, *Halorubrum*, *Halobacteriales*, *Haloterrigena*, *Natronomonas*, *Harlarchaeaum*, *Halorarcula*, *Halonotius*, and *Halorussus*), opening a new path for further investigation [123].

Even though several studies have been performed to identify the archaeal gut microbiome, there is still much unknown about its direct implication in health and disease. The interactions or effects archaea might have on other gut microbiota communities or the host immune system are also unknown. These new findings suggest that archaea have been underestimated in previous microbiota studies and might play an essential role in healthy and pathological conditions. Expanding these studies to a larger number of patients and different geographic areas will provide us with information to discern the healthy archaea microbiota from the ones causing diseases.

## 6. Discussion

In recent years, the significant function of non-classical gut microbiota homeostasis and disease etiology is slowly being discovered. Although several new species have been identified, the cultivation of these species is still challenging. Not being able to isolate these microorganisms makes them difficult to characterize; therefore, designing experiments to study their implication in intestinal health and disease is challenging.

In this context, the impact of helminths, bacteriophages, fungi, and archaea as an essential part of the intestinal microbiota should be further explored since most of their functions in the gut are still unknown. However, this cannot be achieved without considering their interactions with the bacterial community and human gut function. The selection of microbiota may depend on the host’s mucus, mucin, and glycan composition [143,144]. To survive and colonize this hostile environment, gut microorganisms need to develop mechanisms to obtain nutrients and thrive against competition, leading to the synthesis of specific proteins, enzymes, or metabolites. The study of these mechanisms opens a great new research area for the gut microbiota at the molecular level.

Intestinal development during early life mainly depends on environmental stimuli, where bacteria play critical functions by colonizing different segments of the intestinal tract at different times [145]. However, the involvement of non-bacterial microorganisms in intestinal development during early life and their implication on adult health remains unexplored. In this context, fungal species would be present at a much higher diversity during the first months of life compared with later periods, contrasting bacterial diversity [146]. A similar trend in virome diversity occurs during the first years of life [147,148] in helminths [149] and archaea [150], where they are stabilized later in life. Although changes in abundance and diversity of these non-bacterial microorganisms have been detected during early life, their role in intestinal defense development and as critical factors in developing a healthy adult-like microbiota remains uncharted. More studies are needed to state the role of the other microbiota in gut colonization during early life in the same way as the plethora of studies that have identified the function of intestinal bacteria and how they shape the intestine during early life [151].

In this review, we describe the implication of bacterial viruses in the gut microbiota, but the presence of viruses that target other microbiota communities and even human cells must also be addressed.

The unconventional microorganisms of the gut, outlined in this review, are involved in various intestinal diseases, but they can also be developed as attractive therapeutic agents to improve certain GI symptoms. Innovative studies support the idea that some bacteriophages would be helpful in modulating non-beneficial bacteria that can induce the development of some GI diseases, such as UC, CD, or invasive adherent *Escherichia coli* (IAEC) [152]. Nevertheless, the impact on other microbiota communities should be studied in parallel. Even though several works have demonstrated that the use of non-traditional microorganisms in intestinal maladies has improved gastrointestinal symptoms and mitigated intestinal inflammation, their intracellular mechanisms remain unclear. In addition, the effect of specific species of helminths, bacteriophages, fungi, and archaea in intestinal disorders should be well described so that specific components from these organisms could be used as potential therapies against intestinal diseases.

For a more comprehensive study of these gut organisms and their roles in intestinal homeostasis, recent works indicate the necessity of expanding available databases for reliable detection. In addition, the unification of detection methods would be indispensable to ensure a consistent and comparable identification between studies all over the world [153].

In recent years, the use of artificial intelligence (AI), including deep machine learning, to understand biological questions has grown in popularity and become more accessible to the research community. Microbiome data are high-dimensional, sparse, and compositional, requesting specific analysis. We can benefit from the interesting discoveries provided by AI technologies [154], allowing us to find patterns we could not identify directly.

## Figures and Tables

**Figure 1 life-13-01765-f001:**
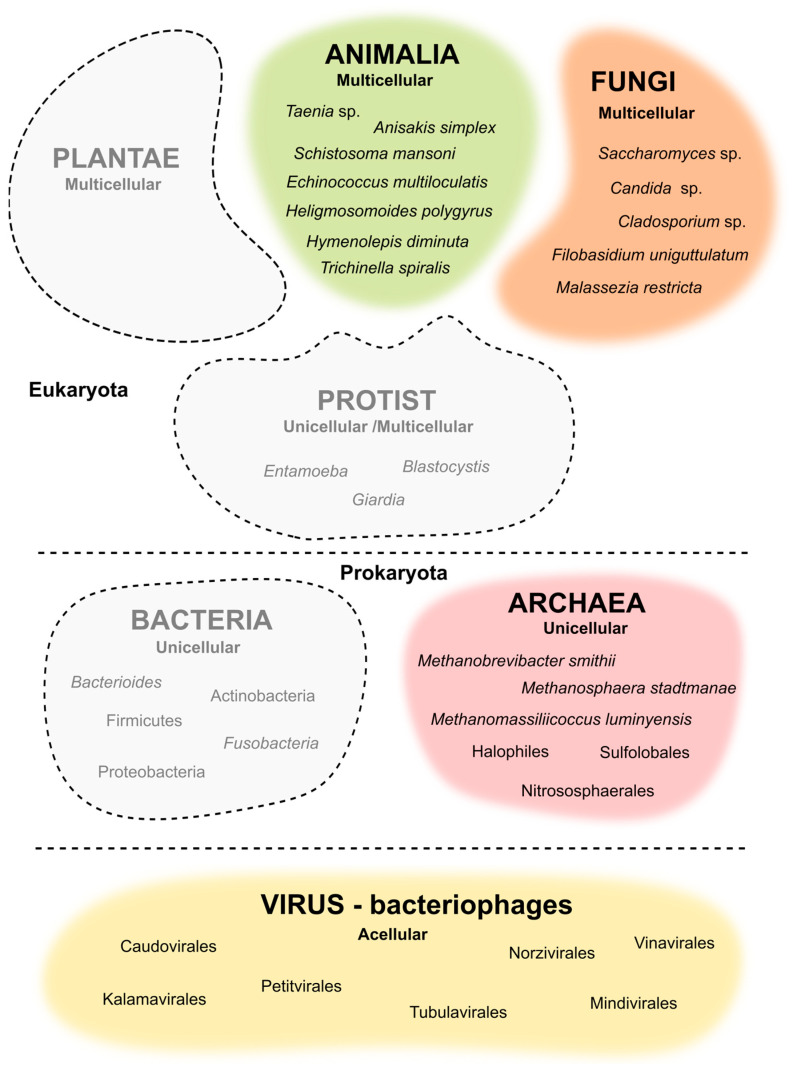
Representation of the underrated microorganism of the human gut microbiota.

**Table 1 life-13-01765-t001:** Most relevant phyla in gut helminths.

Phylum	Species Example	Symptoms
Cestodes	*Taenia saginata*	Usually do not cause symptoms, but infected patients can experience non-specific symptoms such as abdominal discomfort, nausea, vomiting, diarrhea, and weight loss, among others.
	*Taenia solium*	
Nematodes	*Anisakis simplex*	Non-specific symptoms include abdominal pain and general discomfort with uncommon cases of extra-gastrointestinal symptoms such as allergy with angioedema, urticaria, and anaphylaxis.
Trematodes	*Schistosoma mansoni*	Non-specific symptoms include abdominal pain, enlarged liver as well as hematochezia, and hematuria.

**Table 2 life-13-01765-t002:** Classification of bacteriophages.

Order	Family	Characteristics
**Caudovirales**	*Siphoviridae*	dsDNA, long, noncontractile tails
	*Myoviridae*	dsDNA, contractile tails
	*Podoviridae*	dsDNA, short, stubby tails
**Kalamavirales**	*Tectiviridae*	dsDNA, internal membrane
**Petitvirales**	*Microviridae*	ssDNA, circular, icosahedral
**Tubulavirales**	*Inoviridae*	ssDNA filamentous
**Norzivirales**	*Fiersviridae*	ssRNA, small icosahedral
**Mindivirales**	*Cystoviridae*	dsRNA, segmented, enveloped
**Vinavirales**	*Corticoviridae*	dsDNA, circular, internal membrane
**Unassigned**	*Plasmaviridae*	dsDNA, circular, enveloped

**Table 3 life-13-01765-t003:** Alterations in gut phageome during various gastrointestinal diseases modified from Tiamani et al. [69].

Disease	Changes in Phageome
**Ulcerative colitis**	Increased Caudovirales
**Crohn’s disease**	Lytic phages are replaced with lysogenic phages
**Colorectal cancer**	More diverse phageome
**Coeliac disease**	More Enterobacteria phages, fewer Lactococcus and Streptococcus phages

**Table 4 life-13-01765-t004:** Archaea abundance in CRC.

Enriched Archaea CRC	Depleted Archaea CRC
*Halorubrum tropicale*	*Methanocorpusculum* (2 sp.)
*Halococcus morrhuae*	*Methanosarcina* (3 sp.)
*Halococcus salifodinae*	*Methanobrevibacter* (3 sp.)
*Halovenus aranensis*	*Methanobacterium* (3 sp.)
*Natrinema* sps J7-2	*Methanooccus* (2 sp.)
*Methanothrix soehngenii*	*Methanospaera*
*Methanoculleu marisnigri*

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
