# Peer review of "The Underrated Gut Microbiota Helminths, Bacteriophages, Fungi, and Archaea"

_life, 2023, doi:10.3390/life13081765_

Round 1

Reviewer 1 Report

Page 3. Figure 1: Remove italics from "sp" in species names.

Line 64: Remove the yellow mark.

Helminth groups: I suggest a brief explanation/commentary on the phylum Acanthocephala.

Pages 4 and 5: attention in italics in species names

I strongly suggest that the authors increase the information about intestinal helminth infection and microbiota imbalance. Also, include a brief review of the relationship between helminth infection and bacterial translocation. It is very important in a strong article about the gastrointestinal microbiota.

Author Response

We thank reviewer 1 indications. We have modified the manuscript accordingly and the changes can be visualized in the text in red color.

  1. Page 3. Figure 1: Remove italics from "sp" in species names.
    We thank reviewer 1 indication. Accordingly, we have modified the figure and removed the italics from sp.  

  2. Line 64: Remove the yellow mark.
    We have now removed the yellow mark.

  3. Helminth groups: I suggest a brief explanation/commentary on the phylum Acanthocephala.
    Thank you for the suggestion. We have now included a paragraph describing the phylum Acanthocephala.

  4. Pages 4 and 5: attention in italics in species names
    We have gone through the full manuscript and ensured that families, genus, and species names are in italics.

  5. I strongly suggest that the authors increase the information about intestinal helminth infection and microbiota imbalance. Also, include a brief review of the relationship between helminth infection and bacterial translocation. It is very important in a strong article about the gastrointestinal microbiota.
    Thank you for the suggestion. We have included more information about intestinal helminth infection and microbiota imbalance.

Reviewer 2 Report

This review focuses on the role of helminths, bacteriophages, fungi, and Archaea in the gut ecosystem. Since most studies in the field center on bacteria, this review provides a fresh perspective. In general, the review is comprehensive and clear. However, some paragraphs are poorly organized and tedious and need improvement.

1.    The discussion section of the manuscript merely repeats previous paragraphs without providing much new information. I suggest rewriting it to focus on future research directions in this research field.

2.    In line 191, what are crAssphages or Cross Assembly Phage? The authors need to add a few explanation sentences.

3.    The paragraph in lines 206-288 is poorly organized and hard to follow. The authors need to restructure the sentences to allow a clear logical flow.  

4.    The species names should be in italics.

A few words appear unprofessional for a scientific manuscript. These include: ‘nice’ in line 245, ‘relatively unknown’ in line 298, ‘big’ in line 364, ‘statistically accepted’ in line 419, ‘chief’ in line 458, and ‘an interesting tool’ in line 484.

There are a few grammar mistakes that need to be fixed.

Author Response

This review focuses on the role of helminths, bacteriophages, fungi, and Archaea in the gut ecosystem. Since most studies in the field center on bacteria, this review provides a fresh perspective. In general, the review is comprehensive and clear. However, some paragraphs are poorly organized and tedious and need improvement.
We thank reviewer 2 indications. We have modified the manuscript accordingly and the changes can be visualized in the text in red color.

  1. The discussion section of the manuscript merely repeats previous paragraphs without providing much new information. I suggest rewriting it to focus on future research directions in this research field.
    Thank you for the suggestion. We have now modified the discussion and focused more on future perspectives.

  2. In line 191, what are crAssphages or Cross Assembly Phage? The authors need to add a few explanation sentences.
    We have described in the text what crAssphages or Cross Assembly Phage mean.

  3. The paragraph in lines 206-288 is poorly organized and hard to follow. The authors need to restructure the sentences to allow a clear logical flow. 
    We agree with Reviewer 2 and we have shortened and restructured the phrases in lines 206-288.

  4. The species names should be in italics.
    We have gone through the full manuscript and ensured that families, genus, and species names are in italics.

  5. A few words appear unprofessional for a scientific manuscript. These include: ‘nice’ in line 245, ‘relatively unknown’ in line 298, ‘big’ in line 364, ‘statistically accepted’ in line 419, ‘chief’ in line 458, and ‘an interesting tool’ in line 484. There are a few grammar mistakes that need to be fixed.
    We have gone through the full manuscript, and we have corrected the English and avoided unprofessional words for a scientific manuscript.